# Collaborative Geometry-Aware Multi-Solution Optimizer for Efficient Model Fine-Tuning

**Van-Anh Nguyen**
Department of Data Science and AI
Monash University, Australia
van-anh.nguyen@monash.edu

**Trung Le**
Department of Data Science and AI
Monash University, Australia
trunglm@monash.edu

**Mehrtash Harandi**
Department of Electrical and Computer Systems Engineering
Monash University, Australia
mehrtash.harandi@monash.edu

**Ehsan Abbasnejad**
Department of Data Science and AI
Monash University, Australia
ehsan.abbasnejad@monash.edu

**Thanh-Toan Do**
Department of Data Science and AI
Monash University, Australia
toan.do@monash.edu

**Dinh Phung**
Department of Data Science and AI
Monash University, Australia
dinh.phung@monash.edu

## Abstract

We propose a framework grounded in gradient flow theory and informed by geometric structure that provides multiple diverse solutions for a given task, ensuring collaborative results that enhance performance and adaptability across different tasks. This framework enables flexibility, allowing for efficient task-specific fine-tuning while preserving the knowledge of the pre-trained foundation models. Extensive experiments across transfer learning, few-shot learning, and domain generalization show that our proposed approach consistently outperforms existing Bayesian methods, delivering strong performance with affordable computational overhead and offering a practical solution by updating only a small subset of parameters. The code for our method is at https://github.com/anh-ntv/GAC-MSO

## 1  Introduction

The rapid growth of foundation models, particularly Transformers [41] based architecture, has fundamentally transformed the field of artificial intelligence. Pre-trained on massive datasets, these models have demonstrated extraordinary capabilities to learn rich, contextualized representations has led to state-of-the-art performance across a wide range of applications from natural language processing [14, 6, 43, 50] to computer vision [15, 39, 29, 7, 56].

While these models are powerful, adapting them to specific downstream tasks remains challenging due to their enormous size and computational demands. Moreover, in many real-world scenarios, generating multiple diverse solutions for a task can improve robustness, adaptability, and ensemble performance. However, directly optimizing multiple instances of such large models is computationally

39th Conference on Neural Information Processing Systems (NeurIPS 2025).

infeasible. A practical and efficient way to overcome this problem is to optimize a set of compact auxiliary modules that integrate into the foundation model, while keeping most of the original pretrained parameters unchanged and shared across all solutions.

Multiple compact auxiliary modules have been proposed for parameter-efficient fine-tuning (PEFT) frameworks to effectively adapt foundation models to new tasks by tuning only a small fraction of parameters. Initially, full fine-tuning of these models, which often contain millions or billions of parameters, is computationally expensive, memory-intensive, and impractical when multiple tasks are involved. Additionally, limited task-specific data might require careful regularization to prevent overfitting. To address these challenges, pioneering PEFT techniques such as Adapter [20], prompt tuning [22], LoRA [21], SCT [55], and BitFit [52] have demonstrated promising results by maintaining most pre-trained parameters fixed, thus enhancing both computational efficiency and performance.

Building on the efficiency of PEFT, we propose a framework that generates multiple diverse solutions by optimizing lightweight modules while reusing a shared backbone. This design enables collaborative, robust predictions with minimal overhead, preserving the generalization capability of the pre-trained model. We initiate this process by guiding the posterior toward the target distribution using gradient-based updates. A common approach is Stein Variational Gradient Descent (SVGD) [27], which balances high predictive performance with solution diversity through repulsive interactions in parameter space. However, SVGD ignores the geometric relationships among solutions—specifically, how they align in output space or interact within the loss landscape.

To overcome these limitations, we propose the Geometry-Aware Collaborative Multi-Solution Optimizer (GAC-MSO), a theoretically grounded and tractable framework for generating diverse, high-collaborative solutions. Unlike SVGD, which promotes diversity solely in parameter space via kernel-based repulsion, GAC-MSO also integrates geometric structure [3] and enforces output-space diversity through a divergence term. This leads to efficient solution space exploration while maintaining strong predictive accuracy and calibration, even with limited computational resources.

To summarize, our key contributions in this paper include:

- We propose the Geometry-Aware Collaborative Multi-Solution Optimizer (GAC-MSO), a framework grounded in gradient flow theory [3] over the probability space of models. This formulation enables the incorporation of geometric structure [1, 2] and promotes the generation of diverse yet collaborative solutions.
- We conduct extensive experiments on PEFT across various settings, including model fine-tuning for transfer learning, few-shot learning, and domain generalization. The results demonstrate that our GAC-MSO consistently outperforms baseline methods by a significant margin, highlighting the effectiveness of incorporating geometric structure and promoting diverse yet collaborative solutions.

## 2    Related works

**Parameter Efficiency Parameter Tuning.**    Parameter-efficient fine-tuning (PEFT) has gained attention for adapting large pre-trained models to downstream tasks by minimizing computational costs. Several methods have been developed to achieve this.

**– Adapter tuning.** These methods adapt Transformer-based models by inserting lightweight neural modules into each layer and fine-tuning only these modules, while keeping the core model parameters frozen. These adapters typically adopt a bottleneck architecture comprising two small fully connected (FC) layers [20] and an activation function [9]. In the context of vision tasks, certain methods [51] extend the adapter design by incorporating convolutional layers or Normalizing Flows [46] to better capture spatial patterns and complex feature distributions.

**– Prompt Tuning.** These methods adapt models to new tasks by introducing additional learnable visual prompts, which are either inserted into the backbone or applied as perturbations to existing weights. VPT-Shallow [22] inserts prompts only before the first encoder layer, while VPT-Deep [22] places prompts at each encoder layer for deeper integration.

**– LoRA Tuning.** These methods introduce additional parameters during training or fine-tune specific subsets of the model, while ensuring that these modifications are efficiently integrated into the

backbone architecture to minimize inference overhead. A pioneering approach in this category is LoRA [21], which inserts low-rank decomposition matrices into attention layers and merges them with the original weights at inference time, maintaining both computational efficiency and strong performance. Building on LoRA, several variants have been proposed to further enhance its flexibility and effectiveness. AdaLoRA [54] dynamically adjusts the rank of LoRA modules during training based on their importance, improving parameter efficiency. LoRA-Drop [57] introduces dropout within LoRA modules to regularize training and prevent overfitting, particularly in low-resource scenarios.

**Variational Gradient Descent Approach.** This strategy enables sampling multiple models from the posterior distribution, a central technique in neural network inference often realized through Hamiltonian Monte Carlo (HMC) [31]. Although HMC is effective, it is computationally intensive due to its reliance on full gradient evaluations. To improve scalability, Stochastic Gradient HMC (SGHMC) [10] uses noisy gradient estimates, facilitating efficient exploration of the solution space. Similarly, Stochastic Gradient Langevin Dynamics (SGLD) [47] incorporates Langevin dynamics into a stochastic gradient framework. In contrast, Stein Variational Gradient Descent (SVGD) [27] approximates the posterior using a set of interacting particles. Building on this, [44] enhances SVGD with nonlinear transformations that encourage greater particle diversity, addressing SVGD's tendency to collapse in multimodal settings and improving its ability to learn complex mixture models. Complementarily, [11] proposes a repulsive mechanism for deep ensembles that fosters diverse yet plausible members, resulting in a more faithful approximation of the Bayesian posterior.

## 3 Background

### 3.1 Gradient Flow in Probability Space

**Problem Setting.** We start with the problem setting used throughout this paper. Consider the *target distribution* $p(\boldsymbol{\theta}) \propto \exp\{-\beta\Psi(\boldsymbol{\theta})\}$ over $\mathbb{R}^d$, where $\Psi(\cdot)$ is the energy function, we need to find efficient ways of sampling from this target distribution. It should be noted that this setting can be directly applied to Bayesian inference where the energy function is the empirical loss $\mathcal{L}_S(\boldsymbol{\theta})$ over a training set $S = \{(\boldsymbol{x}_i, y_i)\}_{i=1}^N$, which is defined as

$$\mathcal{L}_S(\boldsymbol{\theta}) = \frac{1}{N}\sum_{i=1}^N l(f(\boldsymbol{x_i}; \boldsymbol{\theta}), y_i),$$

where $f(\boldsymbol{x_i}; \boldsymbol{\theta})$ is the prediction output of the model with the model parameter $\boldsymbol{\theta}$ and $l$ is a loss function.

It is evident that $p(\cdot)$ is the solution of the following optimization problem:

$$\min_{\rho \in \mathcal{P}(\mathbb{R}^d)} \left\{ \mathcal{F}(\rho) := \beta \int \Psi d\rho + \int \log \rho d\rho \right\}, \tag{1}$$

where $\mathcal{P}(\mathbb{R}^d)$ is the space of distributions over $\mathbb{R}^d$ with the Wasserstein distance [3].

The gradient flow of $\mathcal{F}$ in the Wasserstein space [3] is described by:

$$\partial_s \rho_s + \text{div}\left(\rho_s \nabla \frac{\partial \mathcal{F}}{\partial \rho_s}\right) = 0,$$

where $\frac{\partial \mathcal{F}}{\partial \rho_s}$ is the first variation (functional derivative) of $\mathcal{F}$ and $\text{div}$ is the divergence operator.

### 3.2 Stein Variational Gradient Descent

Given the current distribution $\rho_t$ as the time step $t$, our aim is to find the velocity field $v_t = id + \eta u_t$ using the steepest descent direction:

$$u_t = \text{argmin}_u \frac{d}{d\eta} \mathcal{F}((id + \eta u)\#\rho_t)\mid_{\eta=0}, \tag{2}$$

where $\#$ is the transport operator, $id$ is the identity function, and $\eta > 0$ is the step size.

The next distribution solution $\rho_{t+1} = v_t \# \rho_t$ where $v_t = id + \eta u_t$. Moreover, by restricting the velocity $u \in \mathcal{H}_K^d$, where $\mathcal{H}_K$ is the Reproducing Kernel Hilbert Space (RKHS) corresponding to the positive semi-definite kernel $K(\boldsymbol{\theta}, \boldsymbol{\theta}') : \Theta \times \Theta \to \mathbb{R}$, Stein Variational Gradient Descent (SVGD) [27] reaches the closed-form solution for the optimal velocity $v_t$ as

$$v_t \left( \tilde{\boldsymbol{\theta}} \right) = \tilde{\boldsymbol{\theta}} + \eta \mathbb{E}_{\boldsymbol{\theta} \sim \rho_t} \left[ -K \left( \boldsymbol{\theta}, \tilde{\boldsymbol{\theta}} \right) \nabla_{\boldsymbol{\theta}} \Psi \left( \boldsymbol{\theta} \right) + \nabla_{\boldsymbol{\theta}} K \left( \boldsymbol{\theta}, \tilde{\boldsymbol{\theta}} \right) \right].$$

# 4 Collaborative Multi-Solution Optimizers

Given the current solution $\rho_t$, to find the next solution $\rho_{t+1}$, we use the proximal operator as follows:

$$\min_{\rho \in \mathcal{P}(\mathbb{R}^d)} \{ \mathcal{F}(\rho) + d(\rho, \rho_t) \}, \tag{3}$$

where $d(\rho, \rho_t)$ is a proximal operator, which is defined below.

## 4.1 Proximal Operator

We define the divergence $d(\rho, \rho_t)$ as

$$d(\rho, \rho_t) = \mathbb{E}_{\boldsymbol{\theta}' \sim \rho, \boldsymbol{\theta} \sim \rho_t} \left[ \mathbb{E}_{\boldsymbol{x}} \left[ KL \left( f \left( \boldsymbol{x}; \boldsymbol{\theta}' \right), f \left( \boldsymbol{x}; \boldsymbol{\theta} \right) \right) + KL \left( f \left( \boldsymbol{x}; \boldsymbol{\theta} \right), f \left( \boldsymbol{x}; \boldsymbol{\theta}' \right) \right) \right] \right], \tag{4}$$

where $KL$ is the Kullback-Leibler (KL) divergence.

In the following lemma, we approximate $d(\rho, \rho_t)$, exposing the geometry around the current solution $\rho_t$. All proof in our theory development can be found in Appendix B in the supplementary material.

**Lemma 4.1.** *The divergence $d(\rho, \rho_t)$ in Eq. (4) can be approximated as*

$$d(\rho, \rho_t) \approx \mathbb{E}_{\boldsymbol{\theta}' \sim \rho, \boldsymbol{\theta} \sim \rho_t} \left[ (\boldsymbol{\theta}' - \boldsymbol{\theta})^\top H(\boldsymbol{\theta}) (\boldsymbol{\theta}' - \boldsymbol{\theta}) \right],$$

*where $H(\boldsymbol{\theta}) = \mathbb{E}_{\boldsymbol{x}} \left[ \mathbb{E}_y \left[ \nabla_{\boldsymbol{\theta}} \log f_y(\boldsymbol{x}; \boldsymbol{\theta}) \nabla_{\boldsymbol{\theta}} \log f_y(\boldsymbol{x}; \boldsymbol{\theta})^\top \right] \right]$ and $f_y(\boldsymbol{x}; \boldsymbol{\theta})$ is the $y$-th prediction output in the prediction probability vector $f(\boldsymbol{x}; \boldsymbol{\theta})$.*

## 4.2 Theory Development

We denote the gradient flow of the optimization problem (OP) in (3) as $(\rho_s)_{s \geq t}$ that satisfies the continuity equation [38] (Chapter 4, Page 110):

$$\partial_s \rho_s + \mathrm{div}(\rho_s v_s) = 0, \forall s \geq t,$$

where $\mathrm{div}$ is the divergence operator and $v_s$ is the velocity field.

Let $f \in T_{\rho_t} \mathcal{M}$ with $\mathcal{M} = \mathcal{P}(\mathbb{R}^d)$ be the perturbation function (*i.e.*, describing how $\rho_t$ changes over time) on the tangent space $T_{\rho_t} \mathcal{M}$ such that

$$f + \mathrm{div}(\rho_t v_t) = 0, \tag{5}$$

which further implies $f = \partial_t \rho_t$.

Thus, for a small step size $\eta > 0$, it follows $f \approx \frac{\rho_{t+\eta} - \rho_t}{\eta}$, which leads to $\rho_{t+\eta} \approx \rho_t + \eta f$. We further derive

$$\mathcal{F}(\rho_{t+\eta}) - \mathcal{F}(\rho_t) \approx \mathcal{F}(\rho_t + \eta f) - \mathcal{F}(\rho_t) \approx \left\langle \eta f, \frac{\partial \mathcal{F}(\rho_t)}{\partial \rho_t} \right\rangle \tag{6}$$

$$= \eta \int \frac{\partial \mathcal{F}(\rho_t)}{\partial \rho_t}(\boldsymbol{\theta}) f(\boldsymbol{\theta}) d\boldsymbol{\theta} \overset{(1)}{=} -\eta \int \frac{\partial \mathcal{F}(\rho_t)}{\partial \rho_t}(\boldsymbol{\theta}) \mathrm{div}(\rho_t(\boldsymbol{\theta}) v_t(\boldsymbol{\theta})) d\boldsymbol{\theta} \tag{7}$$

$$\overset{(2)}{=} \eta \int \left\langle \nabla_{\boldsymbol{\theta}} \frac{\partial \mathcal{F}(\rho_t)}{\partial \rho_t}(\boldsymbol{\theta}), v_t(\boldsymbol{\theta}) \right\rangle \rho_t(\boldsymbol{\theta}) d\boldsymbol{\theta}, \tag{8}$$

where $\overset{(1)}{=}$ is due to Eq. (5) and $\overset{(2)}{=}$ is due to the integral by part.

Noting that $v_t \# \rho_t \approx \rho_{t+\eta}$ by the definition of the velocity field, and relating this to the proximal operator in (3), we arrive at

$$\min_{v_t} \{ \mathcal{F}(v_t \# \rho_t) - \mathcal{F}(\rho_t) + d(\rho, \rho_t) \},$$

which can be reformulated into due to (8) and Lemma 4.1

$$\min_{v_t} \left\{ \eta \int \left\langle \nabla \frac{\partial \mathcal{F}(\rho_t)}{\partial \rho_t}(\boldsymbol{\theta}), v_t(\boldsymbol{\theta}) \right\rangle \rho_t(\boldsymbol{\theta}) d\boldsymbol{\theta} + \mathbb{E}_{\boldsymbol{\theta} \sim \rho_t} \left[ \Delta \boldsymbol{\theta}^\top H(\boldsymbol{\theta}) \Delta \boldsymbol{\theta} \right] \right\}. \tag{9}$$

where $\Delta \boldsymbol{\theta} = v_t(\boldsymbol{\theta}) - \boldsymbol{\theta}$.

Moreover, Theorem 4.2 characterizes the optimal solution of OP in (9), which involves the geometry of the particles sampled from $\rho_t$.

**Theorem 4.2.** *The OP in (9) receives the following optimal solution*

$$v_t^* \left( \tilde{\boldsymbol{\theta}} \right) = \tilde{\boldsymbol{\theta}} - \eta H \left( \tilde{\boldsymbol{\theta}} \right)^{-1} \nabla \frac{\partial \mathcal{F}(\rho_t)}{\partial \rho_t} \left( \tilde{\boldsymbol{\theta}} \right), \tag{10}$$

*where* $H\left( \tilde{\boldsymbol{\theta}} \right) = \mathbb{E}_{\boldsymbol{x}} \left[ \mathbb{E}_y \left[ \nabla_{\boldsymbol{\theta}} \log f_y \left( \boldsymbol{x}; \tilde{\boldsymbol{\theta}} \right) \nabla_{\boldsymbol{\theta}} \log f_y \left( \boldsymbol{x}; \tilde{\boldsymbol{\theta}} \right)^\top \right] \right].$

It should be noted that although the update formula in Theorem 4.2 has a closed form, it is intractable because $\nabla \frac{\partial \mathcal{F}(\rho_t)}{\partial \rho_t} \left( \tilde{\boldsymbol{\theta}} \right) = \beta \nabla \Psi \left( \tilde{\boldsymbol{\theta}} \right) + \nabla \log \rho_t \left( \tilde{\boldsymbol{\theta}} \right)$ is *intractable* due to the term $\nabla \log \rho_t \left( \tilde{\boldsymbol{\theta}} \right)$. In what follows, we present how to estimate this term to obtain a tractable solution.

**Tractable Solution.** To develop a tractable solution, we first notice that $\tilde{v}_t^* \left( \tilde{\boldsymbol{\theta}} \right) = \tilde{\boldsymbol{\theta}} - \eta \nabla \frac{\partial \mathcal{F}(\rho_t)}{\partial \rho_t} \left( \tilde{\boldsymbol{\theta}} \right) = \tilde{\boldsymbol{\theta}} + \eta \tilde{u}_t^*(\tilde{\boldsymbol{\theta}})$ is the velocity so that $\rho_{t+\eta} = \tilde{v}_t^* \# \rho_t$ minimizes $\mathcal{F}(\rho) - \mathcal{F}(\rho_t)$ in a vicinity of $\rho_t$. To find the *optimal increment* $\tilde{u}_t(\tilde{\boldsymbol{\theta}}) = -\nabla \frac{\partial \mathcal{F}(\rho_t)}{\partial \rho_t} \left( \tilde{\boldsymbol{\theta}} \right)$, we seek the steepest descent direction as in Eq. (2). To this end, we strengthen $\mathcal{F}(\rho)$ by adding the divergence term and then replacing $\int \log \rho d\rho$ by a similar term using the convolution operation inspired by [8].

Inspired by [8], to make smooth the entropy function, we redefine $\mathcal{F}(\rho)$ as

$$\mathcal{F}(\rho) = \beta \int \Psi d\rho + \int \log (K * \rho) d\rho, \tag{11}$$

where $K * \rho(\boldsymbol{\theta}) = \int K(\boldsymbol{\theta}, \boldsymbol{\theta}') \rho(\boldsymbol{\theta}') d\boldsymbol{\theta}'$ is the convolution operation using the kernel $K$, which aims to make the entropy function smooth.

Moreover, given the current solution $\rho_t$ and the velocity $\tilde{v}_t = id + \eta \tilde{u}_t$, we define the *divergence term* as

$$\mathcal{L}_{div}(\tilde{u}_t, \eta) = \int l_{\text{div}} \left( \boldsymbol{\theta}_{1:M}^{[\tilde{v}_t]} \right) \prod_{m=1}^M d\rho^{[\tilde{v}_t]} \left( \boldsymbol{\theta}_m^{[\tilde{v}_t]} \right) \tag{12}$$

$$= \int l_{\text{div}} \left( [\boldsymbol{\theta}_m + \eta \tilde{u}_t(\boldsymbol{\theta}_m)]_{m=1}^M \right) \prod_{m=1}^M \rho_t(\boldsymbol{\theta_m}) d\boldsymbol{\theta}_{1:M},$$

where $l_{\text{div}}(\boldsymbol{\theta}_{1:M})$ is the loss that encourages the particles $\boldsymbol{\theta}_{1:M}$ more diverge and $\rho^{[\tilde{v}_t]} = \tilde{v}_t \# \rho_t$.

Conceptually, by minimizing $\mathcal{L}_{div}(\tilde{u}_t, \eta)$ in (12), we aim to learn a velocity $\tilde{v}_t = id + \eta \tilde{u}_t$ in such a way that $\boldsymbol{\theta}_{1:M}^{[\tilde{v}_t]} \sim \rho^{[\tilde{v}_t]} = \tilde{v}_t \# \rho_t$ are encouraged to diverge. Eventually, given the current solution $\rho_t$, we learn the velocity $\tilde{v}_t = id + \eta \tilde{u}_t$ to minimize

$$\mathcal{G}(\tilde{u}_t, \eta) = \mathcal{F}\left( \rho^{[\tilde{v}_t]} \right) + \alpha \mathcal{L}_{div}(\tilde{u}_t, \eta), \tag{13}$$

where $\alpha > 0$ is a trade-off parameter and $\rho^{[\tilde{v}_t]} = \tilde{v}_t \# \rho_t$.

In particular, we aim to find an optimal velocity $\tilde{v}_t$ that simultaneously minimizes $\mathcal{F}(\rho)$ and pushes the diverge particles. In the following theorem, we characterize the steepest descent direction.

**Theorem 4.3.** *The steepest descent direction has the following form:* $\nabla_\eta \mathcal{G}\left(\tilde{u}_t, \eta\right)\big|_{\eta=0} = \langle h, \tilde{u}_t \rangle$, *where* $\langle .,. \rangle$ *is the dot product on* $\mathcal{H}_K^d$ *and*

$$h\left(\cdot\right) = \mathbb{E}_{\boldsymbol{\theta} \sim \rho_t}\left[\beta\nabla\Psi\left(\boldsymbol{\theta}\right)K\left(\boldsymbol{\theta},.\right) - \frac{\mathbb{E}_{\boldsymbol{\theta}' \sim \rho_t}\left[K\left(\boldsymbol{\theta},\boldsymbol{\theta}'\right)\nabla K\left(\boldsymbol{\theta},.\right)\right]}{\mathbb{E}_{\boldsymbol{\theta}' \sim \rho_t}\left[K\left(\boldsymbol{\theta},\boldsymbol{\theta}'\right)\right]}\right]$$
$$+ \alpha\mathbb{E}_{\boldsymbol{\theta}_{1:M} \sim \rho_t}\left[\sum_{m=1}^M \nabla_{\boldsymbol{\theta}_m} l_{div}\left(\boldsymbol{\theta}_{1:M}\right)K\left(\boldsymbol{\theta}_m,.\right)\right].$$

Furthermore, using the first-order Taylor expansion, we obtain

$$\mathcal{G}\left(\tilde{u}_t, \eta\right) = \mathcal{G}\left(\tilde{u}_t, 0\right) + \eta\nabla_\eta\mathcal{G}\left(\tilde{u}_t, \eta\right)\big|_{\eta=0} + O(\eta^2).$$

Therefore, by restricting $\tilde{u}_t$ in the ball of radius $\langle h, h \rangle$ inside $\mathcal{H}_K^d$, we yield $\tilde{u}_t^* = -h$, hence $\tilde{v}_t^* = id + \eta\tilde{u}_t^* = id - \eta h$. Finally, referring to (10), we reach

$$v_t^*\left(\tilde{\boldsymbol{\theta}}\right) = \tilde{\boldsymbol{\theta}} - \eta H\left(\tilde{\boldsymbol{\theta}}\right)^{-1}\nabla\frac{\partial\mathcal{F}\left(\rho_t\right)}{\partial\rho_t}\left(\tilde{\boldsymbol{\theta}}\right) = \tilde{\boldsymbol{\theta}} + \eta H\left(\tilde{\boldsymbol{\theta}}\right)^{-1}\tilde{u}_t^*\left(\tilde{\boldsymbol{\theta}}\right)$$
$$= \tilde{\boldsymbol{\theta}} - \eta H\left(\tilde{\boldsymbol{\theta}}\right)^{-1}h\left(\tilde{\boldsymbol{\theta}}\right).$$

**Practical Method.** In what follows, we present the practical implementation of our method. Similar to SVGD [27], we maintain a set of $M$ particle models, denoted by $\boldsymbol{\theta}_{1:M}$. For implementation convenience, we use the same number of particles as in Eq. (12). To estimate $H(\tilde{\boldsymbol{\theta}})^{-1}$, we approximate it using only its diagonal elements. Furthermore, we compute this estimate using a moving average of $H(\tilde{\boldsymbol{\theta}})$ accumulated from past to current iterations.

$$m_t\left(\tilde{\boldsymbol{\theta}}\right) = \gamma\text{diag}\left(\mathbb{E}_{(\boldsymbol{x},y) \sim \boldsymbol{B}_t}\left[\nabla_{\tilde{\boldsymbol{\theta}}}\log f_y\left(\boldsymbol{x};\tilde{\boldsymbol{\theta}}\right)\nabla_{\tilde{\boldsymbol{\theta}}}\log f_y\left(\boldsymbol{x};\tilde{\boldsymbol{\theta}}\right)^\top\right]\right) + (1-\gamma)m_{t-1}\left(\tilde{\boldsymbol{\theta}}\right),$$

where $\gamma \in [0;1]$ is a momentum decay and $\boldsymbol{B}_t$ is the current mini-batch of $(\boldsymbol{x}, y)$ at the current iteration.

Moreover, each particle $\boldsymbol{\theta}^t$ (i.e., $\boldsymbol{\theta}_{1:M}^t$) is updated as follows:

$$\boldsymbol{\theta}^{t+1} = \boldsymbol{\theta}^t + \frac{\eta}{\left[m_t\left(\boldsymbol{\theta}^t\right) + \epsilon\right]M}\sum_{m=1}^M\left[-\beta\nabla\Psi\left(\boldsymbol{\theta}_m^t\right)K\left(\boldsymbol{\theta}_m^t, \boldsymbol{\theta}^t\right) + \right.$$
$$\left.\frac{\sum_{m'}\left[K\left(\boldsymbol{\theta}_m^t, \boldsymbol{\theta}_{m'}^t\right)\nabla K\left(\boldsymbol{\theta}_m^t, \boldsymbol{\theta}^t\right)\right]}{\sum_{m'}\left[K\left(\boldsymbol{\theta}_m^t, \boldsymbol{\theta}_{m'}^t\right)\right]} - \frac{\alpha}{M}\sum_{m=1}^M\nabla_{\boldsymbol{\theta}_m}l_{\text{div}}\left(\boldsymbol{\theta}_{1:M}^t\right)K\left(\boldsymbol{\theta}_m^t, \boldsymbol{\theta}^t\right)\right], \quad (14)$$

where $\boldsymbol{\theta}^t$ represents a particle model at the $t$-th iteration. The training algorithm is presented in Supplementary.

**Divergence Term.** Our framework facilitates the integration of diverse computational terms to promote both collaboration and diversity among particle models. To illustrate its effectiveness, we introduce a specific term designed to encourage the particle models to generate diverse output predictions, ultimately enhancing the performance of the final ensemble.

We now describe the formulation of the divergence loss $l_{\text{div}}(\boldsymbol{\theta}_{1:M}; \boldsymbol{x}, y)$. For a given data point $(\boldsymbol{x}, y) \in S$, let $f(\boldsymbol{x}; \boldsymbol{\theta}_i)$ denote the predicted probability distribution produced by the particle model $\boldsymbol{\theta}_i$. Define $f_{-y}(\boldsymbol{x}; \boldsymbol{\theta}_i)$ (abbreviated as $f_{-y}^i$ when context permits) as the *non-maximal* prediction vector obtained by removing the ground-truth class $y$ from the prediction. Following the approach in [35], we encourage the non-maximal predictions $f_{-y}^i$ (for $i = 1, \ldots, C$, where $C$ is the number of classes) to be mutually dissimilar, while simultaneously promoting the confidence in the ground-truth predictions $f_y^i$. Drawing inspiration from Determinantal Point Processes (DPP) theory [25], we define the ensemble diversity as:

$$l_{\text{div}}\left(\boldsymbol{\theta}_{1:M}; \boldsymbol{x}, y\right) = -\log\left(\det\left(\left[\tilde{f}_{-y}^i\right]_{i \in [C]}^\top\left[\tilde{f}_{-y}^i\right]_{i \in [C]}\right)\right),$$

where $\tilde{f}^i_{-y} = \frac{f^i_{-y}}{\|f^i_{-y}\|}$ and $\left[\tilde{f}^i_{-y}\right]_{i \in [C]} \in \mathbb{R}^{(C-1) \times K}$ where $\left[C\right] = \left\{1, \ldots, C\right\}$.

Moreover, according to the matrix theory [4],

$$\det\left(\left[\tilde{f}^i_{-y}\right]^\top_{i \in [C\}} \left[\tilde{f}^i_{-y}\right]_{i \in [C]}\right) = \mathrm{Vol}^2\left(\left[\tilde{f}^i_{-y}\right]_{i \in [C]}\right),$$

where $\mathrm{Vol}\left(\left[\tilde{f}^i_{-y}\right]_{i \in [C]}\right)$ specifies the volume spanned the vectors in $\left[\tilde{f}^i_{-y}\right]_{i \in [C]}$, indicating that we aim to maximize the diversity of the non-maximal predictions by maximally increasing their spanned volume.

**Model Fine-tuning with Parameter Efficiency.**   We focus on the fine-tuning problem, where a pre-trained model, denoted as $\Phi$, is provided, and the goal is to identify the optimal parameters $\theta = \Phi + \Delta$, with $\Delta$ representing an additional component. Various parameter-efficient fine-tuning (PEFT) methods, such as LoRA [21], Adapters [20], or prompt-tuning [22] have been developed to achieve this objective and have demonstrated remarkable performance compared to the conventional full fine-tuning. Since $\Delta$ is typically a much smaller component than the complete model in PEFT methods, we can conveniently maintain and learn an empirical distribution over several light-weight components $\Delta$, making our approach feasible.

## 5   Experiments

In this section, we conduct extensive experiments across various settings to validate the effectiveness of our proposed method: Image classification Benchmark, Domain generalization setting, and Few-shot Learning. Each experiment is repeated with three random seeds, and the mean accuracy is reported.

Detail of the experimental setting is presented in Appendix A, which includes the backbone, how to set up multiple particles, the kernel function, and trade-off parameters.

### 5.1   Image classification

**VTAB-1k dataset** [53] consists of 19 distinct datasets, which are grouped into three categories: Natural, Specialized, and Structured. Each dataset contains only 1,000 images for training, making the task challenging due to the limited amount of data. Additionally, the images show significant variation in data distribution across the datasets, further complicating the learning process.

We conduct experiments using four particles for our GAC-MSO and all baselines, except full fine-tuning (FFT), AdamW, and SAM, for which we use a single particle consistent with standard LoRA-based fine-tuning of foundation models. Each particle is randomly initialized at the start.

It's important to note that using four particles increases the total number of trainable parameters by a factor of four compared to single-particle methods. However, more parameters do not necessarily lead to improved performance and can sometimes even degrade it. As shown in Table 1, most baseline methods with multi-solution settings perform worse compared to the single-solution setting methods. This may be due to inefficient model scaling or the trade-off between learning diverse solutions and optimizing individual performance. Despite these challenges, our GAC-MSO method outperforms all baselines, achieving the highest average accuracy with a notable 2.3% improvement.

Additionally, we evaluate all methods using the Expected Calibration Error (ECE), which measures how well the predicted probabilities align with actual outcomes. The results, presented in Table 2, show that GAC-MSO achieves a comparable ECE score to SAM under single-particle settings and outperforms other SAM-based methods, such as SADA-JEM [49] and SA-BNN [32], in the multi-particle scenario. SAM-based approaches are known for producing solutions that lie in flatter regions of the loss landscape, correlating with better generalization. Additionally, our method achieves better ECE performance compared to SVGD [28], which records the best ECE score among the other multi-solution baselines.

**FGVC dataset.** The FGVC benchmark comprises five fine-grained datasets for visual classification tasks: CUB-100-2011 [42], NABirds [40], Oxford Flowers [34], Stanford Dogs [12], and Stanford Cars [17]. Each dataset contains between 1,000 and 21,000 images for training, offering a diverse

Table 1: VTAB-1K results evaluated on Top-1 accuracy. All methods are applied to finetune the same set of LoRA parameters on ViT-B/16 pre-trained with ImageNet-21K dataset.

| | Natural | | | | | | | Specialized | | | | Structured | | | | | | | | |
|---|---|---|---|---|---|---|---|---|---|---|---|---|---|---|---|---|---|---|---|---|
| Method | CIFAR100 | Caltech101 | DTD | Flower102 | Pets | SVHN | Sun397 | Camelyon | EuroSAT | Resisc45 | Retinopathy | Clevr-Count | Clevr-Dist | DMLab | KITTI | dSpr-Loc | dSpr-Ori | sNORB-Azi | sNORB-Ele | AVG |
| **Single solution setting** | | | | | | | | | | | | | | | | | | | | |
| FFT [22] | 68.9 | 87.7 | 64.3 | 97.2 | 86.9 | 87.4 | 38.8 | 79.7 | 95.7 | 84.2 | 73.9 | 56.3 | 58.6 | 41.7 | 65.5 | 57.5 | 46.7 | 25.7 | 29.1 | 65.6 |
| AdamW [21] | 67.1 | 90.7 | 68.9 | 98.1 | 90.1 | 84.5 | 54.2 | 84.1 | 94.9 | 84.4 | 73.6 | **82.9** | 69.2 | 49.8 | 78.5 | 75.7 | 47.1 | 31.0 | 44.0 | 72.0 |
| SAM [16] | 72.7 | 90.3 | 71.4 | 99.0 | 90.2 | 84.4 | 52.4 | 82.0 | 92.6 | 84.1 | 74.0 | 76.7 | 68.3 | 47.9 | 74.3 | 71.6 | 43.4 | 26.9 | 39.1 | 70.5 |
| **Multi-solution setting** | | | | | | | | | | | | | | | | | | | | |
| DeepEns [26] | 69.1 | 88.9 | 67.7 | 98.9 | 90.7 | 85.1 | 54.5 | 82.6 | 94.8 | 82.7 | 75.3 | 46.6 | 47.1 | 47.4 | 68.2 | 71.1 | 36.6 | 30.1 | 35.6 | 67.0 |
| BayesTune [23] | 67.2 | 91.7 | 69.5 | 99.0 | 90.7 | 86.4 | 54.7 | 84.9 | 95.3 | 84.1 | 75.1 | 82.8 | 68.9 | 49.7 | 79.3 | 74.3 | 46.6 | 30.3 | 42.8 | 72.2 |
| SGLD [48] | 68.7 | 91.0 | 67.0 | 98.6 | 89.3 | 83.0 | 51.6 | 81.2 | 93.7 | 83.2 | 76.4 | 80.0 | 70.1 | 48.2 | 76.2 | 71.1 | 39.3 | 31.2 | 38.4 | 70.4 |
| SADA-JEM [49] | 70.3 | 91.9 | 70.2 | 98.2 | 91.2 | 85.6 | 54.7 | 84.3 | 94.1 | 83.4 | **77.0** | 79.9 | **72.1** | **51.6** | 79.4 | 70.7 | 45.3 | 29.6 | 40.1 | 72.1 |
| SA-BNN [32] | 65.1 | 91.5 | 71.0 | 98.9 | 89.4 | **89.3** | 55.2 | 83.2 | 94.5 | 86.4 | 75.2 | 61.4 | 63.2 | 40.0 | 71.3 | 64.5 | 34.5 | 27.2 | 31.2 | 68.1 |
| SVGD [28] | 71.3 | 90.2 | 71.0 | 98.7 | 90.2 | 84.3 | 52.7 | 83.4 | 93.2 | 86.7 | 75.1 | 75.8 | 70.7 | 49.6 | 79.9 | 69.1 | 41.2 | 30.6 | 33.1 | 70.9 |
| **GAC-MSO (Ours)** | **73.7** | **94.9** | **72.6** | **99.4** | **91.6** | 85.8 | **58.3** | **86.2** | **96.2** | **86.9** | 74.0 | 79.0 | 63.8 | 51.0 | **79.9** | **84.4** | **58.3** | **33.4** | **46.4** | **74.5** |

Table 2: VTAB-1K results evaluated on the Expected Calibration Error (ECE) metric. All methods are applied to fine-tune the same set of LoRA parameters on ViT-B/16 pre-trained with ImageNet-21K dataset.

| | Natural | | | | | | | Specialized | | | | Structured | | | | | | | | |
|---|---|---|---|---|---|---|---|---|---|---|---|---|---|---|---|---|---|---|---|---|
| Method | CIFAR100 | Caltech101 | DTD | Flower102 | Pets | SVHN | Sun397 | Camelyon | EuroSAT | Resisc45 | Retinopathy | Clevr-Count | Clevr-Dist | DMLab | KITTI | dSpr-Loc | dSpr-Ori | sNORB-Azi | sNORB-Ele | AVG |
| **Single solution setting** | | | | | | | | | | | | | | | | | | | | |
| FFT [22] | 0.29 | 0.23 | 0.20 | 0.13 | 0.27 | 0.19 | 0.45 | 0.21 | 0.13 | 0.18 | 0.17 | 0.41 | 0.44 | 0.42 | 0.22 | 0.14 | 0.23 | 0.24 | 0.40 | 0.26 |
| AdamW [21] | 0.38 | 0.19 | 0.18 | 0.05 | 0.09 | 0.10 | **0.14** | 0.11 | 0.09 | 0.12 | 0.11 | **0.12** | 0.23 | 0.34 | 0.18 | 0.14 | 0.21 | 0.18 | 0.31 | 0.17 |
| SAM [16] | 0.21 | 0.25 | 0.20 | 0.11 | **0.12** | 0.15 | **0.14** | 0.17 | 0.16 | 0.14 | **0.09** | **0.12** | **0.17** | 0.24 | 0.16 | 0.21 | **0.19** | 0.13 | 0.16 | **0.16** |
| **Multi-solution setting** | | | | | | | | | | | | | | | | | | | | |
| DeepEns [26] | 0.24 | 0.12 | 0.22 | 0.04 | 0.10 | 0.13 | 0.23 | 0.16 | 0.07 | 0.15 | 0.21 | 0.31 | 0.32 | 0.36 | 0.13 | 0.32 | 0.31 | 0.16 | 0.29 | 0.20 |
| BayesTune [23] | 0.32 | 0.08 | 0.20 | 0.03 | 0.85 | 0.12 | 0.22 | 0.13 | 0.07 | 0.13 | 0.22 | **0.12** | 0.23 | 0.30 | 0.24 | 0.28 | 0.28 | 0.31 | 0.26 | 0.23 |
| SGLD [48] | 0.26 | 0.20 | 0.17 | 0.05 | 0.18 | 0.14 | 0.23 | 0.18 | 0.09 | 0.12 | 0.32 | 0.26 | 0.29 | 0.21 | 0.26 | 0.42 | 0.39 | **0.11** | 0.24 | 0.22 |
| SADA-JEM [49] | 0.22 | 0.11 | 0.20 | 0.05 | 0.13 | 0.16 | 0.18 | 0.15 | 0.21 | 0.23 | 0.26 | 0.19 | 0.20 | 0.25 | 0.27 | 0.35 | 0.20 | 0.14 | **0.13** | 0.19 |
| SA-BNN [32] | 0.22 | 0.08 | 0.19 | 0.15 | 0.12 | 0.12 | 0.24 | 0.13 | 0.06 | 0.12 | 0.18 | 0.14 | 0.21 | 0.22 | 0.24 | 0.25 | 0.41 | 0.46 | 0.34 | 0.20 |
| SVGD [28] | 0.20 | 0.13 | 0.19 | 0.04 | 0.16 | **0.09** | 0.20 | 0.15 | 0.11 | 0.13 | 0.12 | 0.17 | 0.21 | 0.30 | 0.18 | 0.21 | 0.25 | 0.24 | 0.26 | 0.18 |
| GAC-MSO (Ours) | **0.14** | **0.03** | **0.16** | **0.00** | **0.06** | 0.11 | 0.15 | **0.12** | **0.03** | **0.08** | 0.18 | 0.16 | 0.29 | 0.38 | **0.05** | **0.09** | 0.25 | 0.41 | 0.38 | **0.16** |

range of challenges for fine-grained image recognition. Detailed results and experimental setup are placed in Appendix A.

The results demonstrate that our GAC-MSO method achieves notable improvements in both accuracy and ECE score. In particular, GAC-MSO significantly outperforms the SVGD approach on calibration, achieving an ECE score of 0.05 compared to 0.14 on SVGD. This highlights the effectiveness of our approach not only in enhancing predictive performance but also in improving model confidence and trustworthiness in fine-grained classification tasks.

## 5.2 Few-shot learning

In this section, we extend our analysis to a few-shot learning setting by varying the number of training samples (shots) per class across 1, 2, 4, 8, and 16. We evaluate performance on five fine-grained datasets: FGVC-Aircraft [30], Oxford-Pets [36], Food-101 [5], Stanford Cars [24], and Oxford-Flowers102 [33].

We adopt the same experimental setup as described in Section 2 for standard image classification, using four particles for GAC-MSO, SVGD [28], and Deep Ensemble methods, and a single particle for AdamW. The results for each dataset are shown in Figure 1, where our GAC-MSO achieves the highest accuracy across most of the shot settings compared to the baselines. The detailed results, including accuracy and ECE scores, are provided in Appendix A. Overall, GAC-MSO also demonstrates a notable improvement in calibration performance, achieving lower ECE scores than competing methods.

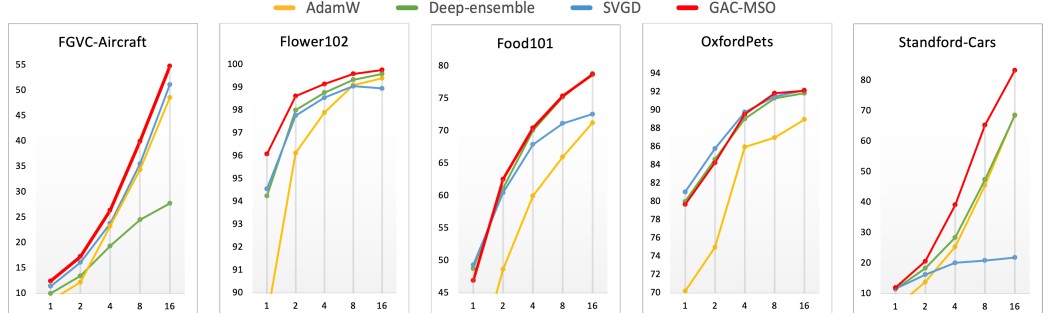

Figure 1: Accuracy on the few-shot benchmark FGVC. The x-axis represents the number of shots (training samples per class) in this setting. Evaluation of ECE scores is provided in Appendix.

Table 3: Top-1 accuracy on domain generalization experiments. All models are fine-tuned with a subset ImageNet-1K dataset and tested on five datasets.

| Method | Accuracy | | | | | ECE | | | | |
| | Source | Target | | | | Source | Target | | | |
| | ImageNet | -Sketch | -V2 | -A | -R | ImageNet | -Sketch | -V2 | -A | -R |
| --- | --- | --- | --- | --- | --- | --- | --- | --- | --- | --- |
| **Single-solution setting** | | | | | | | | | | |
| AdamW [21] | 70.8 | 20.0 | 59.3 | 6.9 | 23.3 | - | - | - | - | - |
| **Multi-solution setting** | | | | | | | | | | |
| Deep-ensemble [26] | 79.4 | 36.2 | 68.9 | 17.6 | 33.9 | 0.069 | **0.028** | **0.041** | 0.130 | **0.034** |
| SVGD [28] | 77.4 | 36.6 | 67.3 | 17.1 | 35.1 | 0.500 | 0.253 | 0.435 | **0.088** | 0.232 |
| GAC-MSO (Ours) | **79.6** | **37.3** | **69.1** | **19.6** | **35.8** | **0.058** | 0.049 | 0.044 | 0.123 | 0.049 |

## 5.3 Domain generalization

We analyze the robustness of our method in practical scenarios where domain shift [58] is unavoidable. In this setting, the model is fine-tuned on a subset of the ImageNet-1K dataset [13], which includes 16 samples per class. After fine-tuning, we test the model on three widely used validation sets derived from ImageNet-1K: the original ImageNet-1K validation set, ImageNet-V2 [37], and ImageNet-Sketch [45]. Additionally, we also include two challenging benchmarks: ImageNet-A [19], which consists of naturally adversarial samples, and ImageNet-R [18], which contains artistic and abstract renditions of ImageNet classes. As shown in Table 3, our GAC-MSO method consistently achieves higher accuracy than all baseline methods across all test sets, including both mild (Sketch and V2) and extreme (Adversarial and Rendition) domain shifts. The improvement gap is notable on the more challenging ImageNet-A and ImageNet-R datasets. Additionally, GAC-MSO maintains a comparable or better ECE score, indicating that the predictions remain well-calibrated even under out-of-distribution conditions.

## 5.4 Effectiveness of geometry-aware and divergency term

In this section, we analyze the effectiveness of two key components: the geometry term, which encourages diversity in the model space by leveraging geometric structure, and the divergence term, which promotes diversity in predictions within the output space. Results are presented in Table 4. Incorporating the geometry term leads to a significant performance improvement compared to models without it, highlighting the benefit of modeling relationships in parameter space. Furthermore, adding the divergence term provides additional gains, demonstrating its effectiveness in enhancing ensemble diversity and improving predictive performance.

## 5.5 Additional experiments on trade-off $\alpha$ of divergence term and number of particles

Detailed experiments and results are presented in the Supplementary.

Table 4: Results on the VTAB-1K Dataset. We report the average performance across the three task groups: natural, specialized, and structured.

| Geometry | Divergence | Natural | Specialized | Structured | Average |
|:---:|:---:|:---:|:---:|:---:|:---:|
| | | 79.77 | 84.60 | 56.25 | 73.54 |
| x | | 82.13 | 85.73 | 60.96 | 76.27 |
| x | x | **82.32** | **85.83** | **62.03** | **76.72** |

## 6 Conclusion and Limitation

In this work, we have addressed the challenges of adapting large foundation models to downstream tasks, particularly when diverse solutions are needed for improved robustness and ensemble performance. We introduced the Geometry-Aware Collaborative Multi-Solution Optimizer (GAC-MSO), a novel framework that leverages parameter-efficient fine-tuning (PEFT) by optimizing lightweight modules while sharing a common backbone. Grounded in gradient flow theory and geometric structure, GAC-MSO promotes diversity not only in parameter space but also in output behavior. Our extensive experimental evaluation across transfer learning, few-shot learning, and domain generalization demonstrates that GAC-MSO significantly outperforms existing baseline methods, providing strong predictive performance with affordable computational cost. These results highlight the potential of GAC-MSO for efficient and effective adaptation of foundation models in resource-constrained settings.

**Acknowledgment**

Trung Le, Mehrtash Harandi, and Dinh Phung were supported by the ARC Discovery Project grants DP230101176 and DP250100262. Trung Le and Mehrtash Harandi were also supported by the Air Force Office of Scientific Research under award number FA9550-23-S-0001.

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
