# OpenReview forum: "Geometry-Aware Collaborative Multi-Solutions Optimizer for Model Fine-Tuning with Parameter Efficiency"
_NeurIPS.cc/2025/Conference — NeurIPS 2025 poster_

### Official Review · Reviewer_x89M · 2025-07-02

**Clarity:** 2
**Significance:** 2
**Originality:** 3
**Rating:** 4
**Confidence:** 2

**Summary:**

This paper proposes the Geometry-Aware Collaborative Multi-Solution Optimizer (GAC-MSO), a gradient flow-based framework that leverages geometric structure to generate diverse yet collaborative solutions for machine learning tasks. Unlike traditional methods like SVGD, which rely solely on parameter-space diversity, GAC-MSO incorporate geometric insights and enforces output-space diversity through a divergence term, enhancing exploration while maintaining predictive accuracy and calibration. Experriments in transfer learning, few-shot learning, and domain generalization show that the proposed approach outperforms existing methods with affordable computational overhead.

**Questions:**

1. How does the number of particles affect ensemble diversity, individual model performance, and computational cost?

**Ethical Concerns:**

["NO or VERY MINOR ethics concerns only"]

**Final Justification:**

As the computational cost analysis has been provided, I have raised my score to 4. But I remain uncertain about my evaluation of this submission.

**Limitations:**

The authors have not discussed the limitations of the proposed framework in the Conclusion and Limitation section.

**Quality:**

3

**Strengths And Weaknesses:**

## Strengths
1. This work offers conceptual innovation by combining SVGD with geometry-aware optimization, backed by a theoretical foundation in gradient flow theory and geometric structure. The proposed divergence term promote diversity in both parameter and output spaces.
2. Experiments are performed across various settings (Image classification, Domain generalization, and fewshot learning) to validate the effectiveness of the proposed method. Ablation studies on the geometry term and the divergence term further confirm the design's effectivenss.

## Weaknesses
1. The proposed method using four particles increases the total number of trainable parameters by a factor of four compared to standard single-particle methods, which may greatly increase computational overhead and reduce practical usefulness. However, there appears to be no analysis or results on computational costs.

---

> ### Author Rebuttal · Authors · 2025-07-31
>
> Dear reviewer,
>
> We thank the reviewer for your valuable the time to review our work. Below, we address the point you have raised.
>
> ---
>
> **Training and inference time** When $M > 1$, the training process involves the loss function $l_{div}$, which requires calculating the determinant of the prediction matrix, and the kernel function $K(., .)$, which involves calculating distances in the model or latent space. As a result, increasing M leads to an exponential increase in training time. However, during inference, since the loss and kernel functions are not involved, the shared architecture among particles can be done in parallel. Therefore, increasing M does not significantly impact the inference time. In practice, the number of particles is selected based on the available system resources and the expected performance. We consider the trade-off between performance and training time when increasing the number of solutions M as our limitation.
>
> _Table 1: Training and inference time (s) on 1 epoch_
> | #num solutions | Training | Inference |
> |-----------------|----------|-----------|
> | 1               | 6.0      | 1.25      |
> | 2               | 13.2     | 1.68      |
> | 3               | 20.4     | 2.83      |
> | 4               | 27.6     | 3.42      |
>
> ---
>
> We agree that increasing the number of particles leads to a linear increase in training time. However, this is a common characteristic shared by particle-based sampling methods (e.g., SVGD and SGLD) as well as ensemble approaches.
>
> We kindly ask the reviewer to consider the theoretical and technical contributions of our work. Our study focuses on particle-based sampling and variational gradient approaches, including well-established methods such as Stochastic Gradient Langevin Dynamics (SGLD) and Stein Variational Gradient Descent (SVGD), which are widely used in Bayesian inference (e.g., Bayesian Neural Networks). Specifically, we consider the gradient flow in the probability space that connects the prior and posterior distributions. We develop theoretical foundations to endow this gradient flow with a well-defined geometry and introduce a divergence term that encourages collaborative yet diverse particle solutions. In our experiments, we compare our method against relevant baselines in Bayesian inference and particle sampling, demonstrating its significant improvement over state-of-the-art methods.
>
> &nbsp;

---

> > ### Author Response · Authors · 2025-08-05
> > **Feedback to our rebuttal**
> >
> > Dear reviewer x89M
> >
> > Thank you for taking the time to read and acknowledge our rebuttal. As the discussion period has been extended for another 48 hours, we would like to kindly follow up to see if our response has adequately addressed your original concerns, or if there are any remaining questions you’d like us to clarify, especially regarding the additional empirical results.
> >
> > We’re happy to elaborate further if needed and remain fully available during the discussion period. We sincerely appreciate your thoughtful feedback, and we hope our response has been helpful.

---

> > > ### Author Response · Authors · 2025-08-06
> > > **Your feedback is appreciate**
> > >
> > > Dear reviewer x89M,
> > >
> > > The author-reviewer discussion is concluding soon. We acknowledge your comment regarding the increase in training time with more particle models. However, this is an inherent characteristic of both ensemble and particle sampling approaches (e.g., SGLD, SVGD). By leveraging lightweight PEFT components, our method remains scalable to large-scale datasets. We have also reported the training times as the number of particles increases. We hope you to consider our theoretical and methodological contributions in your final evaluation. Thank you for your time and effort in reviewing our paper.

---

> > ### Comment · Reviewer_x89M · 2025-08-07
> >
> > I thank the authors for their response, which has addressed my concern regarding the lack of computational cost analysis. I encourage the authors to include detailed computational cost results in the paper. In addition, the "Conclusion and Limitation" section should explicitly discuss the limitations of the work. This issue was raised by all reviewers, but it does not appear to have been seriously addressed in the rebuttal.

---

> ### Author Response · Authors · 2025-08-07
> **The limitation of our work**
>
> Dear Reviewer,
>
> Thanks for your reminder.
>
> We have mentioned the limitation of our work when discussing training and inference time in the rebuttal, where we stated: **"We consider the trade-off between performance and training time when increasing the number of solutions (i.e., $M$) as the limitation of our work."** This limitation is also a common characteristic shared by particle-based sampling methods as well as ensemble approaches. We will carefully discuss this limitation in the revised version.

---

> > ### Comment · Reviewer_x89M · 2025-08-08
> >
> > Thanks for the response. As the computational cost analysis has been provided, I will raise my score to 4.

---

> > > ### Author Response · Authors · 2025-08-08
> > >
> > > We appreciate the reviewer for their consideration.

---

> > > > ### Author Response · Authors · 2025-08-09
> > > > **Thank you!**
> > > >
> > > > The discussion period is coming to a close. We sincerely thank the reviewer for taking the time to review our paper and for providing valuable feedback.

---

### Official Review · Reviewer_dwXi · 2025-07-02

**Clarity:** 3
**Significance:** 3
**Originality:** 3
**Rating:** 5
**Confidence:** 2

**Summary:**

The authors propose a learning framework grounded in gradient flow theory which leverages geometric structure that is supposed to enhance performance and adaptability across different tasks. The authors conduct experiments across transfer learning, few-shot learning, and domain generalization against several existing Bayesian methods.

**Questions:**

The authors take ideas from 8 and 25, but they eventually do not compare against these methods. Are these methods not applicable in this scenario?

**Ethical Concerns:**

["NO or VERY MINOR ethics concerns only"]

**Final Justification:**

I had only a simple question and the authors addressed it to my satisfaction. Given my low expertise, I carefully checked the other reviews. In my understanding, there are no reason that require me to alter my rating.

**Limitations:**

the authors did not discuss limitations, but I can also not think of any. so might be ok?

**Quality:**

3

**Strengths And Weaknesses:**

**Strengths**
- The authors support their method by extensive theoretical grounding. The authors proof several theorems and derive their method on existing work [8, 25]
- Comparison against many baselines and a wide range of datasets and problem settings.
- Ablations investigating the impact of different aspects of the method.

Overall, the paper is well-written and the authors provide strong support for their claims.

**Weaknesses**
I can't think of any

---

> ### Author Rebuttal · Authors · 2025-07-31
>
> Dear Reviewer,
>
> We sincerely thank you for taking the time to review our work. Below, we provide a response to the questions you raised.
>
> &nbsp;
>
> **Q1: Idea from [8] and [25]**
>
> Thanks for this comment. The paper [8] introduces a general family of a Blob approach for diffusion processes. In our paper, we leverage this Blob approach to our framework. Moreover, drawing inspiration from Determinantal Point Processes (DPP) theory [25], we develop our divergence term.
>
>
> &nbsp;
>
> **Q2: Training and inference time** When $M > 1$, the training process involves the loss function $l_{div}$, which requires calculating the determinant of the prediction matrix, and the kernel function $K(., .)$, which involves calculating distances in the model or latent space. As a result, increasing M leads to an exponential increase in training time. However, during inference, since the loss and kernel functions are not involved, the shared architecture among particles can be done in parallel. Therefore, increasing M does not significantly impact the inference time. In practice, the number of particles is selected based on the available system resources and the expected performance. We consider the trade-off between performance and training time when increasing the number of solutions M as our limitation.
>
> _Table 1: Training and inference time (s) on 1 epoch_
> | #num solutions | Training | Inference |
> |-----------------|----------|-----------|
> | 1               | 6.0      | 1.25      |
> | 2               | 13.2     | 1.68      |
> | 3               | 20.4     | 2.83      |
> | 4               | 27.6     | 3.42      |
>
> ---
>
> &nbsp;

---

> > ### Comment · Reviewer_dwXi · 2025-08-03
> >
> > Can you comment on the "Are these methods not applicable in this scenario?" of my question?

---

> ### Author Response · Authors · 2025-08-04
> **Thanks for your question**
>
> We cannot compare to [8,25] because they are only the building-blocks to develop our approach. [8] proposes  a general family of a Blob approach for general diffusion processes. We use that in Eq. (11) in the second term ($K*\rho$) to make smooth the entropy function and come up with the update in Theorem 4.3. Furthermore, [25] discusses how to use determinantal point processes for machine learning in general which we base on to encourage the diversity of the model predictions.

---

### Official Review · Reviewer_jYjW · 2025-07-03

**Clarity:** 2
**Significance:** 3
**Originality:** 3
**Rating:** 4
**Confidence:** 2

**Summary:**

The paper introduces a novel Geometry-Aware Collaborative Multi-Solution Optimizer (GAC-MSO), designed to efficiently fine-tune large pre-trained models for specific downstream tasks. Leveraging gradient flow theory and geometric structure, the authors provide a theoretically grounded optimization solution and a tractable approximation for the method. The proposed method generates diverse yet collaborative solutions by optimizing lightweight fine-tuning modules while capturing the geometric relationships within the model's parameter and output space.  These modules maintain the generalization capabilities of pre-trained backbone models while minimizing computational overhead. Extensive experiments demonstrate that GAC-MSO significantly outperforms existing parameter-efficient fine-tuning (PEFT) approaches across image classification, few-shot learning, and domain generalization tasks.

**Questions:**

- Can you elaborate on the computational overhead introduced by managing multiple particles?
- Could you discuss potential challenges or limitations?

**Ethical Concerns:**

["NO or VERY MINOR ethics concerns only"]

**Final Justification:**

A technically solid paper and the performance of the method have demonstrated great potential. The author addressed the concerns of the computational budget for the method and the ablation studies for the hyperparameters. I've I carefully checked the other reviews and based on my limited expertise in this field, I decided to keep my score.

**Limitations:**

potential limitations are:
- Although the authors present a tractable algorithm, the practical implementation of the proposed Geometry-Aware method might still be complex due to multiple interacting components (e.g., geometry and divergence terms, kernel approximations).
- The effectiveness of the proposed method likely depends significantly on hyperparameters such as the kernel type, number of particles, the divergence regularization parameter $\alpha$. This can lead to extensive hyperparameter tuning.

**Paper Formatting Concerns:**

no formatting issues

**Quality:**

3

**Strengths And Weaknesses:**

Strengths:

- The proposed method accounts for the diversity not just in the parameter space like SVGD, but also among solutions in the output space.
- The proposed approach is theoretically grounded and a tractable algorithm is provided for practice.
- Performance is strong across diverse benchmarks compared to other baselines, surpassing other established PEFT methods.
- The method enhances model robustness through diversified collaborative solutions, maintaining strong performance in domain generalization.

Weaknesses:

- Overall, the paper is a bit hard to follow since the main focus the theories. I think the author can spend a bit more effort on motivation, for instances, explain what are the geometric relationships and why are they important.
- No discussion about limitations

---

> ### Author Rebuttal · Authors · 2025-07-31
>
> We sincerely thank the reviewer for your time to review our work. Below, we provide a response to the questions you raised.
>
> **Q1: The importance of geometry**
>
> Geometry plays a crucial role in our approach. It is derived from the proximal operator in Eq. (3), which incorporates the Fisher information as defined in Eq. (4). From this perspective, the resulting term acts as a geometry-aware regularization. Furthermore, as shown in Eq. (4), the geometry influences the update along each dimension of the model parameters.
>
> &nbsp;
>
> **Q2: Training and inference time** When $M > 1$, the training process involves the loss function $l_{div}$, which requires calculating the determinant of the prediction matrix, and the kernel function $K(., .)$, which involves calculating distances in the model or latent space. As a result, increasing M leads to an exponential increase in training time. However, during inference, since the loss and kernel functions are not involved, the shared architecture among particles can be done in parallel. Therefore, increasing M does not significantly impact the inference time. In practice, the number of particles is selected based on the available system resources and the expected performance. We consider the trade-off between performance and training time when increasing the number of solutions M as our limitation.
>
> _Table 1: Training and inference time (s) on 1 epoch_
> | #num solutions | Training | Inference |
> |-----------------|----------|-----------|
> | 1               | 6.0      | 1.25      |
> | 2               | 13.2     | 1.68      |
> | 3               | 20.4     | 2.83      |
> | 4               | 27.6     | 3.42      |
>
> ---
>
> &nbsp;
>
>
> **Q3: Training algorithm**  Here we present the training algorithm of our GAC-MSO. In our algorithm, we need to compute the loss function $\psi(.)$, $l_{div}(.)$, the kernel function $K(.,.)$, and their respective gradients. In practice, the loss function $\psi(.)$ (which we use as the cross-entropy loss) and $l_{div}(.)$ are calculated using the output logits, and the gradients of these functions are derived simultaneously using the standard backpropagation algorithm. For the kernel $K$, we employ the RBF kernel, which is applied to the model space. The RBF function uses the $l_2$ distance, a computationally simple and efficient measure with a closed-form for its gradient. However, we agree that the overall process adds to the computational cost of the training.
>
>
> **Input:** Initialize particles $\theta_1, \theta_2, ..., \theta_M$, kernel $K(., .)$, momentum $\gamma=0.9$, trade-off $\beta=1$, $\alpha=0.2$, and learning rate $\eta$
>
> **For** each particle $i=1...M$:
>
> **do**
>
> [1]. Compute the empirical estimate of the score function:
>
> $h\left(\theta_i\right)= \sum_{m=1}^{M}\left[\beta\nabla_{\theta_m}\Psi\left(\theta_m\right)K\left(\theta_m,\theta_i\right)-\frac{\sum_{m=1}^{M}\left[K\left(\theta_m,\theta_i\right)\nabla_{\theta_m} K\left(\theta_m,\theta_i\right)\right]}{\sum_{m=1}^{M}K\left(\theta_m,\theta_i\right)}\right]+\alpha\sum_{m=1}^{M}\nabla_{\theta_{m}}l_{\text{div}}\left(\theta_{1:M}\right)K\left(\theta_{m},\theta_i\right).$
>
> $H(\theta_i) = \left(\nabla_{\theta_m}\Psi\left(\theta_m\right) + \nabla_{\theta_{m}}l_{\text{div}}\left(\theta_{1:M}\right)\right)^2$
>
> [3] Using a moving average with momentum:
>
> $m(\theta_i) = \gamma H(\theta_i) + (1-\gamma)m(\theta_i)$
>
> [4] Update the particle using the gradient descent step:
>
> $\theta_i = \theta_i - \frac{\eta}{M(m(\theta_i) + \epsilon)}h(\theta_i)$
>
> **done**
>
> **Output:** Return the updated particles $\theta_1, \theta_2, ..., \theta_M$.
>
> ---
>
> &nbsp;
>
>
> **Q4: The choice of hyperparameters**  In this paper, we formulate the problem in a general form, so that multiple hyperparameters are used, such as the number of particles $M$, the trade-off parameters $\alpha$ and $\beta$ for the loss functions, and the kernel function $K(., .)$. However, in the main experiments, we set $M = 4$, $\alpha = 1$, $\beta = 0.2$, and use an RBF kernel for $K(., .)$ to focus on analyzing the effectiveness of our algorithm in learning diverse solutions with geometry collaboration. The sensitivity of $M$ and $\beta$ is explored in the supplementary material, which demonstrates the effectiveness of each hyperparameter. For the kernel function $K(.,.)$, we use the RBF kernel, as it is a common function to measure the similarity between two points in space. This simple kernel is effective for high-dimensional spaces, such as model or latent spaces.

---

> > ### Comment · Reviewer_jYjW · 2025-08-06
> >
> > I thank the authors for their rebuttal and for providing the additional results. My questions have been addressed.

---

### Official Review · Reviewer_W6Wr · 2025-07-05

**Clarity:** 2
**Significance:** 2
**Originality:** 2
**Rating:** 4
**Confidence:** 4

**Summary:**

The authors introduce Geometry-Aware Collaborative Multi-Solution Optimizer (GAC-MSO), a parameter-efficient framework that learns **multiple** lightweight tuning modules (e.g., LoRA adapters) on top of a frozen foundation backbone.   They derive an update rule that (i) leverages a **geometry term** to keep the particle models well separated in *parameter* space, and (ii) adds an **output-space divergence term** to encourage complementary predictions

Practically, they keep the backbone frozen and train *M* particles (they use *M = 4*) of LoRA modules with the derived update (diagonal Hessian approximation plus moving average).

**Questions:**

Please see the weakness.

**Ethical Concerns:**

["NO or VERY MINOR ethics concerns only"]

**Final Justification:**

I think the paper's contribution is enough. It provides a clear variational derivation, states lemmas and theorems, and sketches proofs


But given that enhancing PEFT robustness is a central motivation and objective of this paper, I believe it is essential to address the more related works in the revision, discuss the compatitbilities with GAC-MSO and compare with baselines which focus on PEFT robustness.

**Limitations:**

No. The authors did not discuss any limitations in the section "Conclusion and Limitation" even though "Limitation" is explicit in the section title. The authors replied "NA" to the limitations in the checklist, meaning "that the paper has no limitation" according to the answer definitions. However, no paper is perfect without limitations, and this paper definitely contains several limitations, such as does the assumptions in the theorems are realistic in real life and the additional GPU memory consumption and flops during inference.

**Quality:**

2

**Strengths And Weaknesses:**

## Strengths
1. The proposed solution is well-motivated based on the existing methods
2. Provides a clear variational derivation, states lemmas and theorems, and sketches proofs
3. The evaluation is broad, including multiple settings and datasets.

## Weakness
1. The goal is to generate diverse solutions for improved robustness and ensemble performance. However, generating diverse solutions requires additional inference cost. With the additional cost, many other approaches are possible. For example, a recent work [1] shows that different PEFT methods generate diverse predictions, and ensembling shows a consistent gain (Figure 4). If GAC-MSO uses 4 particles, it should compare with using 4 different PEFT methods.

2. For robustness, [1] also shows that weight-space ensembles (model merge) can improve robustness (figure 1 (c)).  The authors should compare GAC-MSO with it or discuss the compatibility of GAC-MSO and weight-space ensembles.

3. Additional clarifications are needed to improve the paper. For example
- A figure demonstrating the training and testing pipeline of GAC-MSO would be helpful
- "The final output is then generated by averaging the predictions from all particles". It would be clearer if the author could explain "averaging". It means majority vote, average logits or probability.
- Some citations are confusing. e.g., tabl1. FFT (full fine-tuning)'s citation is VPT. AdamW's citation is LoRA (the whole table is based on LoRa)

[1]  Lessons and Insights from a Unifying Study of Parameter-Efficient Fine-Tuning (PEFT) in Visual Recognition. CVPR 2025.

[2] Robust fine-tuning of zero-shot models. CVPR 2022

---

> ### Author Rebuttal · Authors · 2025-07-31
>
> We appreciate the time and effort you put into reviewing our paper. We truly appreciate your thoughtful feedback and valuable comments, and we have addressed the points you raised below.
>
> **Q1: Comparison with ensemble of multiple PETF methods**
>
>  We thank the reviewer for pointing out the interesting paper [1]. However, we believe that it does not compromise the novelty and technical contributions of our work. Specifically, our study focuses on particle-based sampling or variational gradient approaches, including well-established methods such as Stochastic Gradient Langevin Dynamics (SGLD) and Stein Variational Gradient Descent (SVGD), which are widely used in Bayesian inference (e.g., Bayesian Neural Networks). Technically, we consider the gradient flow in the probability space that connects the prior and posterior distributions. We then develop theoretical foundations to endow this gradient flow with a geometry, along with a divergence term that promotes collaborative yet diverse particle solutions. In our experiments, we compare our method against relevant baselines in Bayesian inference and particle sampling.
>
> Motivated by your suggestion, we experimented using a deep ensemble of four different PEFT components following [1]. For our approach, inspired by [2], we compute the kernel based on model representations—that is, we feed the current mini-batch into four particle models and use their representations to compute the kernel similarity. It is worth noting that in the original results reported in the paper, the kernel was computed based on particle parameters.
>
>
> _Table 1: Comparison with ensemble of multiple PETF methods with GAC-MSO_
> | Method                             | **Natural** | **Specialized** | **Structured** | Average |
> |------------------------------------|-------------|-----------------|----------------|---------|
> | **LoRA, Adapter, VPT-Deep, SSF**   |             |                 |                |         |
> | Deep-ensemble                      | 82.08       | 85.56           | 62.75          | 76.79   |
> | **4 particles LoRA**               |             |                 |                |         |
> | GAC-MSO                            | **82.33**        | **86.40**            | **63.48**           | **77.40**    |
>
> The experimental results show that our approach outperforms the ensemble of four distinct particles, demonstrating the effectiveness of our proposed collaborative and divergent solution.
>
> We also acknowledge the strength of the ensemble approach with four different PEFT particles. This simple strategy provides sufficiently diverse solutions that enhance accuracy. This observation motivates a compelling research question: how can we design a new particle sampling approach in which each particle employs a distinct PEFT type? We leave this promising direction for future work.
>
> [2] Xingchao Liu, Xin Tong, and Qiang Liu. Profiling pareto front with multi-objective stein variational gradient descent. NeurIPS 2021.
>
> ---
>
> &nbsp;
>
> **Q2: Output ensembling**
>
> For the final output, we average the logits before they are passed into the SoftMax function to calculate the accuracy, which is a standard ensembling approach. This will be clearly explained in our revised version.
>
> ---
>
> &nbsp;
>
> **Q3: Training and inference time**
>
> When $M > 1$, the training process involves the loss function $l_{div}$, which requires calculating the determinant of the prediction matrix, and the kernel function $K(., .)$, which involves calculating distances in the model or latent space. As a result, increasing M leads to an exponential increase in training time. However, during inference, since the loss and kernel functions are not involved, the shared architecture among particles can be done in parallel. Therefore, increasing M does not significantly impact the inference time. In practice, the number of particles is selected based on the available system resources and the expected performance. We consider the trade-off between performance and training time when increasing the number of solutions M as our limitation.
>
> _Table 2: Training and inference time (s) on 1 epoch_
> | #num solutions | Training | Inference |
> |-----------------|----------|-----------|
> | 1               | 6.0      | 1.25      |
> | 2               | 13.2     | 1.68      |
> | 3               | 20.4     | 2.83      |
> | 4               | 27.6     | 3.42      |
>
> ---
>
> &nbsp;
>
>
> **Q4: Training algorithm**
>
> Due to NeurIPS policy, we are unable to use external links for figures. Instead, we present the training algorithm of our GAC-MSO here for a clearer understanding.
>
> ---
> **Input:** Initialize particles $\theta_1, \theta_2, ..., \theta_M$, kernel $K(., .)$, momentum $\gamma=0.9$, trade-off $\beta=1$, $\alpha=0.2$, and learning rate $\eta$
>
> **For** each particle $i=1...M$:
>
> **do**
>
> [1]. Compute the empirical estimate of the score function:
>
> $h\left(\theta_i\right)= \sum_{m=1}^{M}\left[\beta\nabla_{\theta_m}\Psi\left(\theta_m\right)K\left(\theta_m,\theta_i\right)-\frac{\sum_{m=1}^{M}\left[K\left(\theta_m,\theta_i\right)\nabla_{\theta_m} K\left(\theta_m,\theta_i\right)\right]}{\sum_{m=1}^{M}K\left(\theta_m,\theta_i\right)}\right]+\alpha\sum_{m=1}^{M}\nabla_{\theta_{m}}l_{\text{div}}\left(\theta_{1:M}\right)K\left(\theta_{m},\theta_i\right).$
>
> $H(\theta_i) = \left(\nabla_{\theta_m}\Psi\left(\theta_m\right) + \nabla_{\theta_{m}}l_{\text{div}}\left(\theta_{1:M}\right)\right)^2$
>
> [3] Using a moving average with momentum:
>
> $m(\theta_i) = \gamma H(\theta_i) + (1-\gamma)m(\theta_i)$
>
> [4] Update the particle using the gradient descent step:
>
> $\theta_i = \theta_i - \frac{\eta}{M(m(\theta_i) + \epsilon)}h(\theta_i)$
>
> **done**
>
> **Output:** Return the updated particles $\theta_1, \theta_2, ..., \theta_M$.
>
> ---
>
> &nbsp;
>
> **Citation confusion** The result of the FFT is taken from paper VPT. And the experiments for single solution using AdamW are taken from the LoRA paper because in this paper, the authors used AdamW to train. We think it will be fair to use the original results.

---

> ### Author Response · Authors · 2025-08-07
> **Additional experiments**
>
> Thank you for your comments and suggestions. In our paper, we report ensemble accuracies and Expected Calibration Error (ECE), which are widely used in prior works on particle sampling and Bayesian Neural Networks. Inspired by your suggestions, we further investigate the robustness of our approach using the relevant methodology introduced in [1].
>
> _Table 3. Comparison between GAC-MSO and WiSE. For WiSE, the number of alpha present for the number of models to ensemble_
> | Method | Source | | Target | | |
> |---|:---:|---:|---:|---:|---:|
> |  |  ImageNet |  -Sketch |  -V2 | -A | -R |
> | **Merge-space WiSE** |  |  |  |  |
> | 4 alphas  | 79.38 | 36.19 | 68.84 | 17.47 | 33.40 |
> | 5 alphas  | 79.40 | 36.20 | 68.89 | 17.49 | 33.40 |
> | 10 alphas | 79.38 | 36.18 | 68.85 | 17.51 | 33.39 |
> | **Multiple-solution** |  |  |  |  |
> | Deep-ensemble (4 particles) | 79.4 | 36.2 | 68.9 | 17.6 | 33.9 |
> | GAC-MSO (4 particles) | **79.6** | **37.3** | **69.10** | **19.6** | **35.8** |
>
> In Table 3, we compare our GAC-MSO (using 4 particles with 4 LoRA models) to both Deep-ensemble (also using 4 particles with 4 LoRA models) and Merge-space WiSE (which trains a single LoRA but ensembles multiple models using different alpha values), as requested by the reviewer. The number of alphas in WiSE corresponds to the number of ensemble models, with the exact alpha values for each setting as follows: 4 alphas using [0.1, 0.4, 0.7, 1]; 5 alphas using [0.1, 0.3, 0.5, 0.7, 0.9]; and 10 alphas using [0.1, 0.2, 0.3, 0.4, 0.5, 0.6, 0.7, 0.8, 0.9, 1]. Our results show that, despite WiSE's implicit flattening of the loss landscape (as the original proposal for full fine-tuning), the outputs from different alpha values lack sufficient diversity to improve robustness when compared to Deep-ensemble and GAC-MSO.

---

> > ### Comment · Reviewer_W6Wr · 2025-08-08
> >
> > Thanks for the detailed response. I have another question. Is GAC-MSO compatible with WiSE? Can we improve GAC-MSO with Wise?

---

> > > ### Comment · Reviewer_W6Wr · 2025-08-09
> > >
> > > I think the paper's contribution is enough. But given that enhancing PEFT robustness is a central motivation and objective of this paper, I believe it is essential to address the related work [1] in the revision and discuss the compatitbilities with GAC-MSO. I will raise the scores accordingly.

---

> > > > ### Author Response · Authors · 2025-08-09
> > > > **Additional experiments with WiSE**
> > > >
> > > > Dear reviewers,
> > > >
> > > > Thank you for your consideration. We would like to report the additional experiments here.
> > > >
> > > > In this study, we explore the use of WiSE with GAC-MSO. All experiments are trained using the GAC-MSO algorithm with 3 particles on 55 epochs.
> > > >
> > > > For the **WiSE for each particle individually** setting, the configuration mirrors the setup in paper [1], with the key difference being that we use 3 particles and train the model with GAC-MSO. As a result, the total number of models in the ensemble is (number of particles * number of alpha) (i.e., 12 models for 4 alpha and 30 models for 10 alpha)
> > > >
> > > > In the **weight-space for all particles** setting, we merge the weights of all particles into one, applying different alpha values for each. This setup is similar to the full fine-tuning application of WiSE, except that we use multiple particles and merge them together.
> > > >
> > > > The results demonstrate that using WiSE effectively amplifies the ensemble results without increasing training time. However, the improvement is heavily dependent on the choice of alpha value.
> > > >
> > > >
> > > > _Table 4. WiSE with GAC-MSO_
> > > > | Method | Num enemble | Source | | Target | | |
> > > > |---|:---:|---:|---:|---:|---:|---:|
> > > > |  |   |  ImageNet |  -Sketch |  -V2 | -A | -R |
> > > > | GAC-MSO | 3 | 75.20 | 36.93 | 65.38 |18.20| **36.53** |
> > > > | **WiSE for each particle individually** |  |  |  |  |
> > > > | 4 alphas | 12  | **75.72** | 37.10 | **65.71** | **18.53** | 36.47 |
> > > > | 10 alphas | 30 | **75.72** | 37.10 | 65.66 | 18.44 | 36.44 |
> > > > | **weight-space for all particles** |  |  |  |  |
> > > > | alpha=[0.1, 0.2, 0.3] | 1 | 75.63 | **37.13** | 65.49 | 18.34 | 36.34 |
> > > > | alpha=[0.1, 0.2, 0.4]| 1   | 75.57 | 37.03 | 65.49 | 18.40 | 36.29 |
> > > > | alpha=[0.1, 0.4, 0.4]  | 1 | 75.29 | 36.94 | 65.49 | 18.14 | 36.37 |

---

> > > > > ### Author Response · Authors · 2025-08-09
> > > > > **Thank you!**
> > > > >
> > > > > The discussion period is coming to a close. We sincerely thank the reviewer for taking the time to review our paper and for providing valuable feedback.

---

> ### Author Response · Authors · 2025-08-08
>
> Dear reviewer,
>
> As the discussion period is coming to an end, we would like to express our appreciation to the reviewer for reviewing our paper and pointing out such interesting work. We would appreciate it if the reviewer could let us know if there are any questions left unresolved, so we can address them promptly before finalizing our rebuttal.

---

### Note · Authors · 2025-08-13

Dear Area Chair and Reviewers,

We sincerely thank the reviewers for their thorough and thoughtful evaluations. We are encouraged that all reviewers acknowledged the novelty and contributions of our work, and confirmed that their concerns were satisfactorily addressed in our rebuttal.
To support the Area Chair’s evaluation, we summarize below the key contributions and the main concerns raised during the review process, along with how we addressed them.

### **Novelty and Key Contributions**

Acknowledged by all reviewers, our proposed GAC-MSO builds on a solid theoretical foundation in gradient flow theory and geometric structure to address the challenge of adapting large foundation models to downstream tasks. Our approach explicitly promotes diversity in both the parameter space and output behavior, enhancing robustness and improving ensemble performance.

### **Concerns Addressed**

1. **Comparison with ensembles of different PEFT methods and WiSE** (raised by Reviewer W6Wr)

* We recognized the relevance of these approaches and conducted additional experiments and detailed discussions comparing GAC-MSO with deep ensembles of multiple PEFT methods and with WiSE under multiple configurations.
* Results show that GAC-MSO consistently outperforms these baselines, and our analysis offers further insight into their respective strengths and limitations.
* We are pleased that this extended comparison fully addressed the reviewer’s concerns.

2. **Training algorithm and computational cost analysis** (raised by Reviewers W6Wr, jYjW, dwXi)

* We provided the training algorithm along with detailed measurements of training and inference times.
* We confirmed the performance–training time trade-off when increasing the number of particles, a well-known characteristic of particle-based sampling methods (e.g., SVGD, SGLD) and ensemble approaches.
* Importantly, increasing the number of particles does not significantly affect the inference time thanks to parallel execution within the shared backbone architecture.

We greatly appreciate the constructive feedback. We remain committed to incorporating these insights—along with the additional experiments and clarifications provided during rebuttal—into the final version of the manuscript.

---

### Decision · Program_Chairs · 2025-09-17

**Decision:**

Accept (poster)

**Comment:**

The paper proposes an optimization approach for fine-tuning a small component of a pre-trained model.  Experiments show performance advantages over competing parameter-efficient fine-tuning strategies.  After the rebuttal and discussion phase, all reviewers lean toward accept, noting solid technical development and sufficient experimental validation.  The Area Chair agrees with the reviewer consensus.